# Comparison of temperature and doping dependence of elastoresistivity near a putative nematic quantum critical point

J. C. Palmstrom [1,2,3,7✉], P. Walmsley[1,2,3], J. A. W. Straquadine [1,2,3], M. E. Sorensen[1,3,4], S. T. Hannahs [5], D. H. Burns[6] & I. R. Fisher [1,2,3✉]

Strong electronic nematic fluctuations have been discovered near optimal doping for several families of Fe-based superconductors, motivating the search for a possible link between these fluctuations, nematic quantum criticality, and high temperature superconductivity. Here we probe a key prediction of quantum criticality, namely power-law dependence of the associated nematic susceptibility as a function of composition and temperature approaching the compositionally tuned putative quantum critical point. To probe the 'bare' quantum critical point requires suppression of the superconducting state, which we achieve by using large magnetic fields, up to 45 T, while performing elastoresistivity measurements to follow the nematic susceptibility. We performed these measurements for the prototypical electron-doped pnictide, $Ba(Fe_{1-x}Co_x)_2As_2$, over a dense comb of dopings. We find that close to the putative quantum critical point, the elastoresistivity appears to obey power-law behavior as a function of composition over almost a decade of variation in composition. Paradoxically, however, we also find that the temperature dependence for compositions close to the critical value cannot be described by a single power law.

[1] Geballe Laboratory for Advanced Materials, Stanford University, Stanford, CA 94305, USA. [2] Department of Applied Physics, Stanford University, Stanford, CA 94305, USA. [3] Stanford Institute for Materials and Energy Sciences, SLAC National Accelerator Laboratory, Menlo Park, CA 94025, USA. [4] Department of Physics, Stanford University, Stanford, CA 94305, USA. [5] National High Magnetic Field Laboratory, Florida State University, Tallahassee, FL 32310, USA. [6] Department of Geological Sciences, Stanford University, Stanford, CA 94305, USA. [7] Present address: National High Magnetic Field Laboratory, Los Alamos, NM 97545, USA. ✉email: jpalmstrom@lanl.gov; irfisher@stanford.edu

A connection between superconductivity and magnetic quantum criticality has been established for a number of heavy fermion systems[1]. Tentative signatures of the effects of possible quantum phase transitions, such as renormalization of the quasiparticle effective mass, have been found for some cuprate superconductors[2,3], but the situation is less clear due to the possible presence and interaction of multiple nearby electronic phases. Compared to cuprates, the situation in the Fe-based materials is much clearer since the symmetry of the ordered phases is well understood and the phase transitions are clearly identified. There is strong evidence for mass renormalization approaching a possible quantum critical point in isovalently substituted $Ba(Fe_{1-x}P_x)_2As_2$[4–7]. For $Ba(Fe_{1-x}Co_x)_2As_2$, recent evidence for power-law scaling of the nematic critical temperature as a function of non-thermal tuning parameters[8] and characteristic scaling of a marginal Fermi liquid found via Raman scattering[9], both point to the existence of a quantum critical point in this system. However, to date, power-law scaling of neither the magnetic nor nematic susceptibility has been observed as a function of composition, and despite suggestive signatures, the universality of quantum criticality has not been established. Indeed, it remains an open question for most Fe-based superconductors whether there is avoided criticality[10–12], or one or two quantum critical points 'hidden' beneath the superconducting dome. The two candidate quantum critical points are a nematic quantum critical point which would have associated rotational symmetry breaking fluctuations and an antiferromagnetic critical point with associated spin-fluctuations. Here, we specifically focus on nematic fluctuations and the variation of the nematic susceptibility upon approach to the associated putative quantum critical point since this is the first of the two possible quantum critical points that are encountered upon approaching the ordered states from the overdoped (tetragonal and non-magnetic) regime. $Ba(Fe_{1-x}Co_x)_2As_2$ was chosen as a representative electron-doped system since the crystal growth of this material is very well controlled, and it is possible to prepare closely spaced compositions spanning the compositionally tuned phase diagram —a key requirement for any test of power-law behavior.

Close to a quantum critical point the susceptibility ($\chi$, where this could be either the magnetic or nematic susceptibility depending on the type of the quantum critical point considered) is anticipated to follow power-law behavior both as a function of doping ($\lim_{T \to 0} \chi \propto |x - x_c|^{-\gamma}$) and temperature ($\lim_{x \to x_c} \chi \propto T^{-\frac{\gamma}{z\nu}}$). Here $T$ is temperature, $x$ is doping, and $x_c$ is the doping at the quantum critical point. $\gamma$ and $z\nu$ are critical exponents that depend on the nature of the critical point. For a nematic quantum critical point in a metal, the system is anticipated to be in a regime where $d + z > 4$, where the effective dimensionality $d$ is already increased due to coupling of the nematic fluctuations to the crystal lattice[13,14]. Consequently, the standard Hertz–Millis approach predicts the mean-field exponent, $\gamma = 1$. Distance from the critical point, both in temperature and in doping, will introduce increasingly large corrections to the power-law scaling.

By symmetry the nematic susceptibility ($\chi_{B_{2g}}$) is related to a specific component of the elastoresistivity tensor[15], $m_{B_{2g}}^{B_{2g}}$ (the linear resistivity response ($\Delta\rho$) to shear strain ($\epsilon_{B_{2g}}$)) by a constant of proportionality ($g_{T,x}$) which can in principle be a function of temperature $T$ and doping $x$,

$$m_{B_{2g}}^{B_{2g}} = \frac{(\frac{\Delta\rho}{\rho_0})_{B_{2g}}}{\epsilon_{B_{2g}}} = g_{T,x}\chi_{B_{2g}} \qquad (1)$$

Here $\rho_0$ is the in-plane resistivity of the unstrained, tetragonal material. In practice we approximate $\rho_0$ with the $\epsilon_{B_{2g}} = 0$ value of the resistivity, $\rho(\epsilon_{B_{2g}} = 0)$. A more detailed and general description of this technique can be found in prior publications[15–18]. Previous measurements of $m_{B_{2g}}^{B_{2g}}$ for underdoped compositions reveal a Curie–Weiss functional form. Since this is the anticipated behavior for $\chi_{B_{2g}}$ approaching a thermally driven nematic phase transition, it was deduced that $g_{T,x}$ did not have an observable temperature dependence for those compositions. Subsequent elastocaloric effect measurements further established a negligible temperature dependence of $g_{T,x}$, i.e. $g_{T,x} \propto g_x$[19]. These measurements also indicate, though, that $g_x$ increases by almost an order of magnitude as $x$ varies from 0 to ≈6% (i.e. approaching $x_c$ from the underdoped side of the phase diagram). The behavior of $g_{T,x}$ for $x > x_c$ has not been established, complicating the interpretation of $m_{B_{2g}}^{B_{2g}}$ and its relation to the nematic susceptibility. Consequently, we have taken an empirical approach, and simply investigate how $m_{B_{2g}}^{B_{2g}}$ varies as a function of $x - x_c$ as $x$ approaches $x_c$ from the overdoped side of the phase diagram.

Measurements close to the putative nematic quantum critical point are further complicated by the presence of superconductivity. Not only does superconductivity preclude resistance measurements, but it also competes with, and induces a back-bending of, the structural transition[20]. Suppressing superconductivity in large magnetic fields removes the competition between superconductivity and the structural transition and permits resistivity measurements to considerably lower temperatures and for compositions much closer to the putative quantum critical point. The elastoresistivity response of $Ba(Fe_{1-x}Co_x)_2As_2$ has a negligible field dependence up to 45 T (Fig. 1a), meaning that large magnetic fields are a small perturbation on the nematic fluctuations.

Elastoresistivity measurements performed in large magnetic fields reveal that the magnitude of $m_{B_{2g}}^{B_{2g}}$ continues to smoothly increase with decreasing temperature in the absence of superconductivity and shows no evidence of saturation for compositions with $x \geq 0.068$. Underdoped samples ($x \lesssim 0.067$) exhibit a tetragonal-to-orthorhombic structural transition which coincides with or precedes a downturn in the elastoresistivity response (Fig. 1b). The tetragonal-to-orthorhombic structural transition is suppressed with doping towards zero temperature near where $m_{B_{2g}}^{B_{2g}}$ is largest (Fig. 2). Assuming that the phase transition remains continuous, the critical doping ($x_c$), i.e. where the transition occurs at zero temperature, is estimated to be $0.067 \pm 0.002$ (see Supplementary Fig. 3 and the Supplementary Note 2). This composition marks the putative nematic quantum critical point in the absence of superconductivity. Formally, since elastoresistivity measurements probe the 'bare' electronic fluctuations, the appropriate critical doping to consider for these measurements is the zero temperature nematic quantum phase transition in the absence of the cooperative effects from the lattice ($x_c^{bare}$). In this work, we make the assumption that $x_c^{bare} \approx x_c$, but reasonable variation in estimates of $x_c$ do not affect any of our conclusions given the uncertainty in all estimates of composition. This is discussed in more detail in Supplementary Note 2.

## Results

**Doping dependence.** The doping dependence of $m_{B_{2g}}^{B_{2g}}$ at 13 K (the lowest temperature where superconductivity can be suppressed for all dopings in 45 T) is shown in Fig. 3. There is an apparent divergence of $m_{B_{2g}}^{B_{2g}}$ upon approach to $x_c$ from the far overdoped side, with a maximum at $x = 0.068$ ($x \approx x_c$). Samples for $x \lesssim 0.067$ are in the ordered phase at this temperature. To look for a power-law dependence we plot the data for samples with

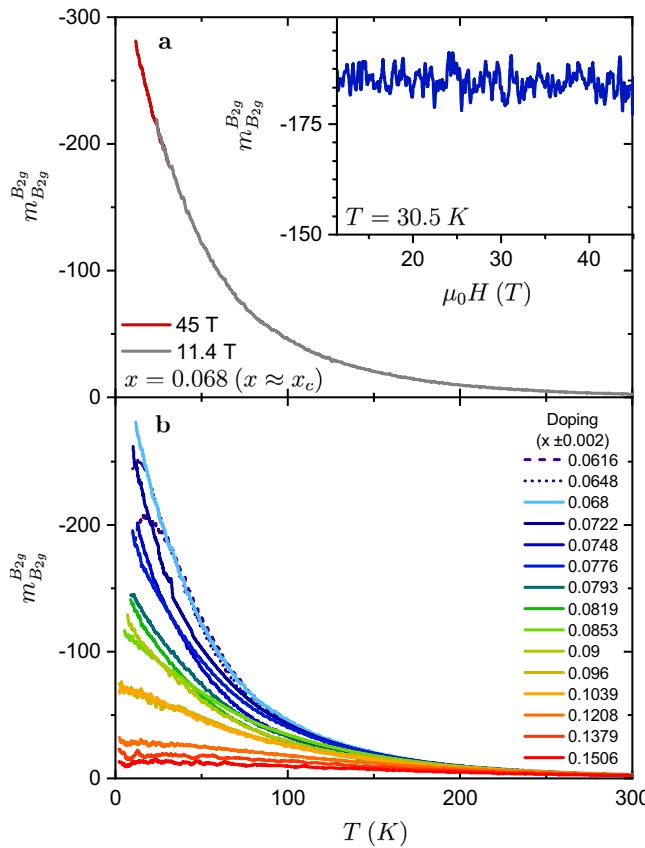

**Fig. 1 The doping and temperature dependence of $m_{B_{2g}}^{B_{2g}}$. a** Temperature dependence of $m_{B_{2g}}^{B_{2g}}$ for the composition closest to $x_c$, Ba(Fe$_{0.932}$Co$_{0.068}$)$_2$As$_2$, in fields of 11.4 and 45 T. Inset shows the absence of any observable field dependence of $m_{B_{2g}}^{B_{2g}}$ above the zero-field superconducting transition. **b** The evolution of $m_{B_{2g}}^{B_{2g}}$ with doping as a function of temperature. Data taken with fields between 0 and 45 T.

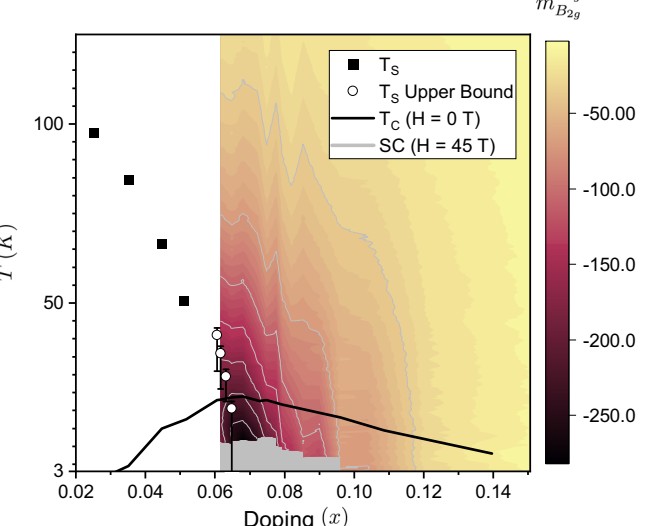

**Fig. 2 Phase diagram of Ba(Fe$_{1-x}$Co$_x$)$_2$As$_2$ overlaid with the doping dependence of $m_{B_{2g}}^{B_{2g}}$ for compositions with $x \geq 0.0616$ (color plot).** The black line is the zero-field superconducting transition and the gray region is the 45 T superconducting dome. The far underdoped structural transition temperatures (black squares) and zero-field superconducting transition temperatures are from J.-H. Chu et al.[27] The white circles represent the onset of the structural transition taken from resistivity measurements at 45 T (see Supplementary Fig. 2 and Supplementary Note 2). Critical temperatures for the magnetic phase transition are not shown.

$x \geq 0.068$ on a logarithmic scale and fit using the functional form, $\log|m_{B_{2g}}^{B_{2g}}| \propto -\phi\log|x - x_c|$. The reduced doping axis, $x - x_c$, spans nearly two decades from 0.001 to 0.0838 with 13 different compositions. Due to finite uncertainty in the measured doping concentration ($\pm 0.002$) there are large errors associated with the reduced composition ($x-x_c$) for the sample closest to the putative critical point ($x = 0.068$) and it is excluded from fits. We performed a linear fit of the data for $x > 0.068$ over a sliding 5-point window (Fig. 3b) using the York computational method to account for $x$ and $y$ errors[21]. For windows that do not include the three most overdoped samples ($x \leq 0.1039$) the extracted slopes agree to within the standard error.

The above analysis indicates that $m_{B_{2g}}^{B_{2g}}$ is consistent with a power-law scaling versus $x - x_c$ for ($0.0722 \leq x \leq 0.1039$) which corresponds to nearly a decade in reduced doping (0.0052–0.0369). We cannot rule out other diverging functional forms, such as a lognormal distribution, but the fact that the observed behavior is at least consistent with a power law is suggestive of critical behavior. The temperature dependence of the extracted exponent, $\phi$, is shown in Fig. 3c. The fitted $\phi$ smoothly increases with decreasing temperature down to the lowest measured temperature (13 K) which corresponds to a value of $\phi(13\text{K}) = 0.72^{+0.18}_{-0.16}$. If $\phi$ continues to smoothly increase, in the limit of $T \rightarrow 0$ K, $\phi(T \rightarrow 0$ K) must be greater than this value. There is a small, temperature independent elastoresistivity response, $m_0$, which is expected to be on the order of the geometric factor. For the range of physically motivated values for $m_0$ the

conclusions drawn here are robust and the extracted $\phi$ at 13 K agree to within error with the $m_0 = 0$ fits shown in Fig. 3.

**Temperature dependence.** Next we return to the temperature dependence of the measured elastoresistivity. For underdoped compositions the temperature dependence of $m_{B_{2g}}^{B_{2g}}$ has been found to follow a Curie–Weiss functional form, $m_{B_{2g}}^{B_{2g}} = \frac{C}{(T-\Theta)} + m_0$[15–18], where $C$ is the Curie constant, $\Theta$ is the Weiss temperature, and $m_0$ is the temperature independent elastoresistivity response. The temperature evolution of $m_{B_{2g}}^{B_{2g}}$ for both the sample closest to the critical doping, $x = 0.068$ ($x \approx x_c$), and a far underdoped sample $x = 0.025$ ($x << x_c$) is shown in Fig. 4 along with the best Curie–Weiss fits and residuals. The Curie–Weiss fit for the $x = 0.068$ sample was performed over the whole temperature range with the best fit parameters $C = -10960 \pm 36$, $\Theta = -20.3 \pm 0.1$, and $m_0 = 39.3 \pm 0.2$. The data and fit for the $x = 0.025$ sample were taken from H.-H. Kuo et al.[18] The fit was performed over a temperature window of 100–205 K with best fit parameters $C = -2706 \pm 32$, $\Theta = 77 \pm 0.8$, and $m_0 = 14.5 \pm 0.8$. The low temperature cutoff is fixed by the structural transition. The fit for the $x = 0.068$ sample not only has an unphysical value for the temperature independent response $m_0$, but the residual clearly has a large systematic temperature dependence above the background measurement noise indicating that the data are not faithfully described by this functional form. In comparison, the residual for the underdoped $x = 0.025$ sample is considerably smaller. Over smaller temperature windows the data for the $x = 0.068$ sample can be well fit by Curie–Weiss, but the data for the low temperature values of $m_{B_{2g}}^{B_{2g}}$ always fall below the divergence expected from Curie–Weiss behavior. This subCurie–Weiss behavior has been previously observed[18,22], however here the measurements are performed over a larger temperature range and on a dense doping series through $x \approx x_c$ where the susceptibility, if

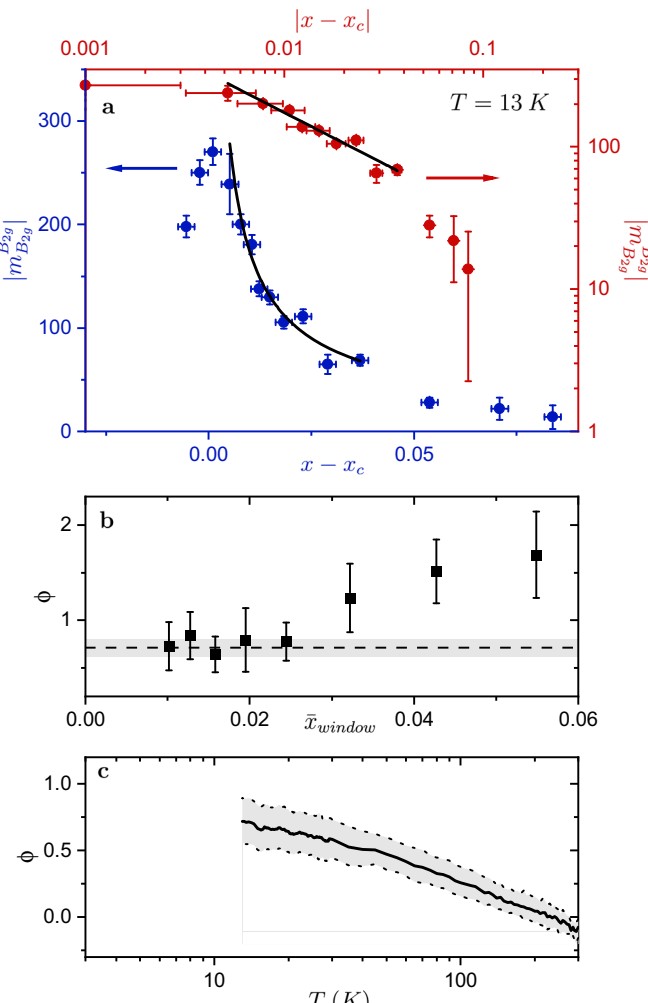

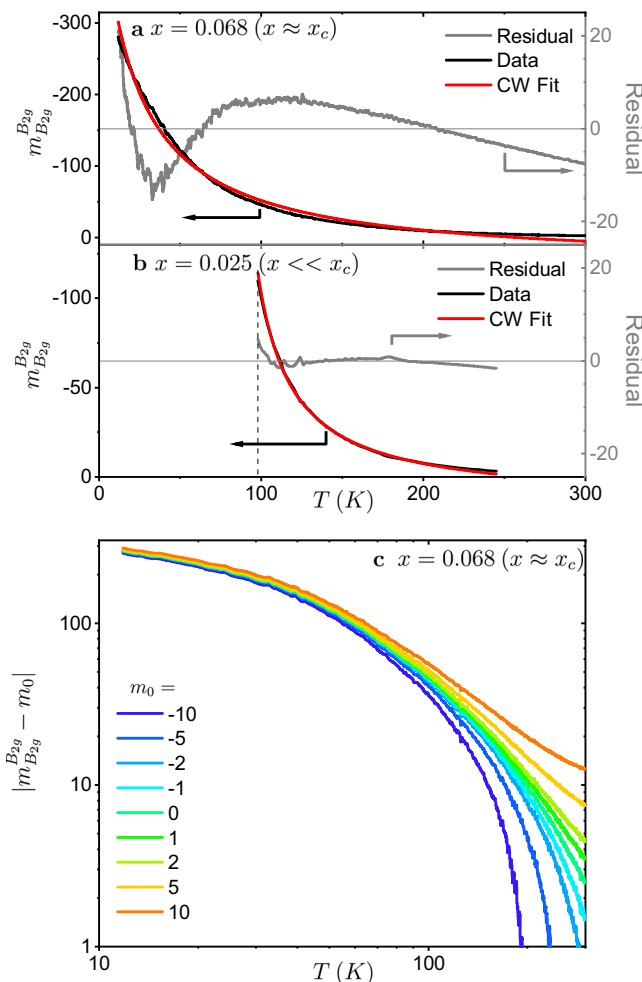

**Fig. 3 Apparent power-law behavior of $|m_{B_{2g}}^{B_{2g}}|$ as a function of $x-x_c$. a** A linear (blue axes) and logarithmic (red axes) plot of $|m_{B_{2g}}^{B_{2g}}|$ vs. $x - x_c$ at 13 K with power-law fit $|m_{B_{2g}}^{B_{2g}}| \propto |x - x_c|^{-\phi}$ (black lines). Error bars include the standard deviation of the measurement in addition to systematic errors. Additional details on included error available in Supplementary Note 1. The fit was performed by fitting a line on the logarithmic plot for $0.0722 \leq x \leq 0.1039$ using the York computational method[21]. **b** The fitted exponent, $\phi$, for fits performed on a sliding 5 point window shown as a function of the average value of $x$ for the window. Overlaid on the plot is the extracted $\phi$ from the fit performed in panel (**a**) (dashed line) and associated standard error (gray region), $\phi = 0.72 \pm 0.09$. Error bars on each data point represent one standard error. Fits that do not include the three most overdoped samples all agree to within the standard error. **c** The measured $\phi$ (black line) as a function of temperature. Error (gray region) includes the standard error of the fits and error associated with uncertainty in the critical doping $x_c$ (see Supplementary Figs. 5–7 and the Supplementary Note 3).

driven by quantum critical fluctuations, is expected to be a power law in the clean limit. Theoretically effects due to weak disorder are predicted to suppress the divergence of the nematic susceptibility upon approach to a nematic quantum critical point[18] which is qualitatively consistent with the observed behavior.

At $x = x_c$ the nematic susceptibility is expected to diverge at zero temperature, i.e. if it was well-described by a Curie–Weiss functional form at the critical doping we expect $\Theta = 0$. To look for power-law behavior in the elastoresistivity with any exponent in temperature we plot the data on a logarithmic scale with a range of physically motivated values for $m_0$ (Fig. 4c). Power-law behavior would result in a linear variation on the logarithmic

**Fig. 4 The temperature dependence of $m_{B_{2g}}^{B_{2g}}$ for $x = 0.068$ ($x \approx x_c$) cannot be described by a simple power law. a** $m_{B_{2g}}^{B_{2g}}$ for $x = 0.068$ (black line), the best Curie–Weiss fit $m_{B_{2g}}^{B_{2g}} \propto \frac{C}{(T-\Theta)} + m_0$ (red line) and the associated residual (gray line). There is a clear temperature dependence in the residual indicating that Curie–Weiss does not fully describe the temperature evolution. **b** $m_{B_{2g}}^{B_{2g}}$ for a far underdoped sample $x = 0.025$ (black line), the best Curie–Weiss fit (red line) and the associated residual (gray line). The data and fit are taken from H.-H. Kuo et al.[18]. This sample has a structural transition at 98 K (dashed line). The magnitude of the residual is small compared to the residual shown in panel (**a**) indicating that Curie–Weiss is a reasonable approximation of the functional form. **c** Logarithmic plot of $|m_{B_{2g}}^{B_{2g}} - m_0|$ vs. temperature for $x = 0.068$ ($x \approx x_c$). No physically motivated value for $m_0$ linearizes the data, demonstrating that $m_{B_{2g}}^{B_{2g}}$ cannot be described by a power law over the whole temperature range.

plot, however, no value for $m_0$ linearizes the data. This indicates that the temperature dependence of the data for the $x = 0.068$ ($x \approx x_c$) sample not only is not described by Curie's law, but in fact cannot by described by any single power law over the entire temperature range measured here. Additional attempted power law fits, including finite $\Theta$ values can be found in Supplementary Fig. 8.

## Discussion

The phenomenology revealed by these high field measurements is of an elastoresistivity that diverges as a function of composition as $x$ approaches $x_c$ from the overdoped side of the phase diagram, with a functional form that is not markedly inconsistent with a simple power law, but with a temperature dependence that very

clearly does not follow a power law. Recalling that the elastoresistivity $m_{B_{2g}}^{B_{2g}} = g_{T,x}\chi_{B_{2g}}$, the observed behavior necessarily reflects the temperature and doping dependence of both of these quantities, $g_{T,x}$ and $\chi_{B_{2g}}$. For underdoped compositions, it has been recently established that the resistivity anisotropy becomes increasingly sensitive to nematic order[23] and nematic fluctuations[19] as $x$ is increased from zero towards $x_c$, so it is certainly not unreasonable to anticipate that $g_{T,x}$ also varies as a function of $x$ on the overdoped side of the phase diagram. Raman scattering measurements reveal that the dynamic nematic susceptibility grows as the composition is tuned towards $x_c$ from the overdoped side of the phase diagram[24], so it is clear that the observed power-law divergence of the elastoresistivity cannot be solely due to doping dependence of $g_{T,x}$. If $g_{T,x}$ does not have a singular $x$-dependence, then the observed power-law behavior of $m_{B_{2g}}^{B_{2g}}$ is suggestive of critical behavior of $\chi_{B_{2g}}$. Of course, such a conclusion can only be tentative while the scaling is observed at relatively high temperatures, for just one decade in reduced composition, and without a more detailed understanding of the doping dependence of $g_{T,x}$. The value of $\phi(13K) = 0.72^{+0.18}_{-0.16}$ that we observe deviates from the anticipated mean-field value of $\gamma = 1$ for clean systems[13,14]. Reasonable extrapolations of $\phi(T)$ to lower $T$ (Fig. 3c), to minimize $T$ dependent corrections to the power-law scaling, are perhaps not inconsistent with this value. However, without measurements to even lower $T$ (requiring yet higher fields to suppress the superconductivity) we cannot rule out other scenarios. In particular, it is possible that the above assumptions do not hold and that $g_{T,x}$ more drastically effects the measured elastoresistivity than anticipated, that the material is not actually in a regime of universal scaling despite signatures seen in other work[8,9], or possibly a different theoretical framework that incorporates disorder effects is necessary to fully capture the phenomenology.

The doping dependence is in marked contrast to the temperature dependence of the elastoresistivity for compositions $x \approx x_c$. For these compositions, it has been established that $g_{T,x}$ does not exhibit a strong temperature dependence[19] and hence the observed non-power-law behavior of $m_{B_{2g}}^{B_{2g}}(T)$ clearly demonstrates that $\chi_{B_{2g}}(T)$ does not follow a power law in temperature, at least down to 13 K (the practical lowest normal-state temperature accessible for elastoresistivity measurements in fields of 45 T close to optimal doping). Recent strain-tuning measurements have clearly demonstrated the presence of quantum critical nematic fluctuations for underdoped compositions[8], implying the existence of a nematic quantum critical point. The deviations from power-law behavior of $\chi_{B_{2g}}$ as a function of temperature might arise from corrections to the scaling relation at high temperatures (i.e. above 13 K). An alternative perspective would be that the system evolves from a high temperature quantum critical regime where the energy scale of disorder is irrelevant, to a lower temperature regime in which disorder becomes increasingly significant. Quenched disorder can lead to rare-region effects which fundamentally change the properties of the system upon approach to the critical point, for instance, $z$ develops a temperature dependence. This has recently been proposed as one possible explanation for the non-power-law behavior seen in similar elastoresistivity measurements[25]. Rare-region effects and qualitatively similar deviations from power-law scaling have also recently been observed in the related material, $FeSe_{1-x}S_x$[26]. The present results point to the need to investigate the associated nematic susceptibility in the absence of superconductivity to even lower temperatures, even closer to the inferred nematic quantum critical point.

## Methods

**Sample growth and characterization.** Bulk single crystal samples were grown by using a self-flux technique[27]. The Co-doping was measured for all material batches using electron microprobe analysis (EMPA). The parent compound $BaFe_2As_2$ and cobalt metal were used for calibration. Doping variation within a sample and within a batch were found to be characterized by a standard deviation of <0.002.

**Sample preparation.** Bulk samples were cleaved into square plates with in-plane dimensions ≥750 μm and out-of-plane dimensions ≤40 μm. The samples were cut such that the edges were parallel to the tetragonal [110] direction. Gold pads were deposited on the corners of the samples using plasma sputtering and an aluminum foil mask. Electrical connection was made by dipping gold wires into EPO-TEK H20E conductive silver epoxy and adhering them onto the gold pads. The resistance of this setup is dominated by the gold wires and typical resistances are ≤3 Ω. This modified Montgomery configuration allows for resistivity measurements simultaneously along the tetragonal [110] and [1$\bar{1}$0] directions.

**Elastoresistivity measurements.** Stress was applied to the samples by gluing them onto piezoelectric stacks (Part No. PSt150/5x5/7 cryo 1, from Piezomechanik GmbH) with Masterbond EP21TCHT-1 epoxy. The samples were glued such that the edges were parallel to the edges of the piezoelectric stack (PZT) and the sample was submerged in epoxy with only a thin layer between the sample and PZT. Two samples were glued onto the front PZT face and a bi-directional resistive strain gauge (Micro-Measurements WK-06-062TT-350) was glued onto the back face of each PZT. The PZT was then mounted such that the applied magnetic field was perpendicular to the $ab$-plane of the samples. Two PZT stacks, with compositions $x = 0.0722$, 0.0853, 0.096, and 0.1208, detached from the probe wall during the experiment. The close 45 T superconducting transition temperatures to nearby compositions suggests that the possible misalignment of field from rotation of these stacks is minimal.

The PZTs were driven from a sine wave generated by a SR860 lock-in amplifier passed through a Tegam 2350 high voltage amplifier. The drive frequency was 23 Hz with an amplitude of 75 $V_{peak}$ for low temperature measurements and 50 $V_{peak}$ for high temperature measurements (typically the cooldown or low field temperature sweeps up to room temperature). Typical temperature sweep rates were 0.7 K/min for low temperature/high field measurements and 3 K/min for temperature sweeps up to 300 K. Current was sourced into the samples and strain gauges by a voltage controlled current source (CS580) which was driven from a sine wave generated by a SR860 lock-in amplifier. The current amplitude was 5 $mA_{RMS}$ and 1 $mA_{RMS}$ through the samples and strain gauges respectively. Typical current frequencies were 30–40 Hz for the samples and 200–400 Hz for the strain gauges. A heating test was performed at 8 K and heating was found to be ≤0.15 K for the maximum PZT drive and sample currents.

AC elastoresistivity measurements[28] were performed by directly locking into the side band using the dual mode of the SR860 lock-in amplifiers. A second SR860 for each channel was used to directly measure the average voltage. The strain gauges were measured through a Wheatstone bridge while the sample voltages were measured directly. A Savitzky–Golay filter with a 1 K window was used to remove background noise. Typically strain was measured along two orientations, parallel and orthogonal to the PZT poling axis. In some instances it was not possible to measure both strain gauges so the average measured Poisson ratio from all runs was used to calculate the overall strain.

All samples were measured in the 45 T Hybrid Magnet in a Helium-4 variable-temperature insert at the National High Magnetic Field Lab except for the two most overdoped samples, $x = 0.1379$ and $x = 0.1506$, which were measured in a 14 T PPMS made by Quantum Design. The measurements on the two overdoped samples were performed with a PZT drive voltage of 50 $V_{peak}$, a temperature sweep rate of 1 K/min, and filtered over a 4 K window.

## Data availability

The datasets generated in this study have been deposited in the Stanford Digital Repository (SDR) database under the Digital Resource Unique IDentifier (DRUID) rh973jh7848.

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

## Acknowledgements

The authors would like to thank R.M. Fernandes, A. Hristov, M. Ikeda, Y. Schattner, S.A. Kivelson, S. Raghu, and Q. Si for helpful discussions. We thank D. Graf and A. Suslov for their assistance with measurements at the National High Magnetic Field Laboratory. EMPA measurements were done at the Stanford Microchemical Analysis Facility. A portion of this work was performed at the National High Magnetic Field Laboratory, which is supported by the National Science Foundation Cooperative Agreement No. DMR-1644779 and the State of Florida. This work was supported by the Department of Energy, Office of Basic Energy Sciences, under Contract No. DE-AC02-76SF00515. J.C.P. was supported by a Gabilan Stanford Graduate Fellowship and a Stanford Lieberman Fellowship. During final revision of the paper, J.C.P. was supported by a Reines Distinguished Postdoctoral Fellowship through the Laboratory Directed Research and Development program of Los Alamos National Laboratory under project number 20200680PRD1.

## Author contributions

J.C.P., P.W., and I.R.F. conceived of the experiment. J.C.P. and P.W. prepared and synthesized the samples. J.C.P. and D.B. performed the EMPA measurements. J.C.P., P.W., J.A.W.S., and M.E.S. performed the measurements in high magnetic fields with assistance from S.T.H. J.C.P. performed the data analysis with guidance from P.W. and I.R.F. J.C.P. and I.R.F. wrote the paper. All authors contributed to editing the manuscript.

## Competing interests

The authors declare no competing interests.
