## [Peer Review File · Nature Communications]

Comparison of temperature and doping dependence of elastoresistivity near a putative nematic quantum critical pointREVIEWER COMMENTS

Reviewer #1 (Remarks to the Author):

I have reviewed the manuscript:

"Comparison of temperature and doping dependence of nematic susceptibility near a putative nematic quantum critical point" by J. C. Palmstrom et al..

This work reports elastoresistivity measurements in order to follow the temperature dependence of the nematic susceptibility in the vicinity of a putative nematic critical point, x_c , in $\text{Ba}_2(\text{Fe}_{1-x}\text{Co}_x)\text{As}_2$. In order to access the putative nematic critical point, the work reports challenging experiments in 45T as well as in 11T to suppress superconductivity using a large number of samples with compositions around that point. This work accesses the low temperature dependence of the nematic susceptibility and assess the power law dependences in the vicinity of the x_c both as a function of temperature and composition. It is found that the temperature dependence of the nematic susceptibility does not follow a power law close to x_c whereas a power law is determined as a function of composition. This study brings interesting insight into the nature of quantum criticality in $\text{Ba}_2(\text{Fe}_{1-x}\text{Co}_x)\text{As}_2$.

Below I suggest additional aspects to be addressed and clarified:

- 1) Fig. 1a shows the nematic susceptibility in magnetic field of 11T and 45T and the curves seem to match. The authors could also add to this graph the zero data nematic susceptibility for the same composition $x \sim 0.068$. It is rather surprising that there is no magnetic field effect on the nematic susceptibility that is related to the anisotropy of in-plane resistivity. The magnetotransport and Hall effect of iron-based superconductors have usually non-linear field dependencies due to the multi-band effects and anisotropic scattering. The authors should comment on how exactly these effects cancel out in the nematic susceptibility and an example in the Supplemental Materials would be useful.
- 2) Fig.1b shows m_{B2g} versus temperature for different composition combining measurements performed in different magnetic fields. The authors should show the related m_{B2g} vs T data in zero field in SM. How the Weiss temperature differs between zero field and the 45T data?
- 3) As the Weiss temperature has been used as an indication of the location of the putative nematic critical point in other studies in different families of iron-based superconductors, it would be useful to have an additional panel in Fig 3 to show the behaviour of the Weiss temperature versus composition and comment on its dependence on composition.
- 4) The findings of this work can be compared to the behaviour of resistivity for the same compositions. The authors could show the resistivity curves in 0T and 45T as separate panels together with their derivative in Figure SM1. This could help to assess the behaviour of resistivity close to x_c , as compared with other studies in the vicinity of a nematic or a magnetic critical point.
- 5) This work identifies a power law dependence between m_{B2g} and the $(x-x_c)^\gamma$ from the overdoped close to x_c at 0.067. What is the behaviour expected from the underdoped side if x_c is the nematic critical point? How this current study compared with the previous findings in 65T published in Phys. Rev. B 100, 125147 (2019)?
- 6) A temperature of $T \sim 13\text{K}$ is used to compare the absolute values of the m_{B2g} as a function of composition the power law dependence on composition in the vicinity of x_c . Can the authors comment on the extrapolated zero temperature values of m_{2g} (0K) versus composition?
- 7) In what way the magnetic field play any role in helping to distinguish between a nematic or magnetic quantum critical point? Can the two potential critical points be evaluated from the resistivity data in 45T?

8) What is the role of disorder in understanding the deviation from Curie-Weiss law at low temperatures? In a previous report, the authors suggested that m_{66} coefficient of optimally doped $\text{BaFe}_2(\text{As}_{0.68}\text{P}_{0.32})_2$ follows a perfect Curie-Weiss temperature dependence from $T = 250$ K (Science 352, 6288 (2016)). For $\text{FeTe}_{0.6}\text{Se}_{0.4}$, the m_{66} coefficient was found to be described by the Curie-Weiss dependence "down to T_c but substantially deviates at temperatures greater than 100 K, possibly related to the loss of quasiparticle coherence as a result of its extremely small Fermi energy". In what way the Curie-Weiss dependence of nematic susceptibility is sensitive to the details of each system, including the size of the Fermi surface, orbital-dependent effects and disorder?

Reviewer #2 (Remarks to the Author):

The authors have experimentally measured and discuss the B_{2g} elastoresistance coefficient (expected to be proportional to the nematic susceptibility) of overdoped $\text{Ba}(\text{Fe},\text{Co})_2\text{As}_2$. Superconductivity has been suppressed in magnetic fields up to 45T. The authors show power-law behavior of the magnitude of $m_{B_{2g}}$ at a constant temperature as a function of composition. They also demonstrate that $m_{B_{2g}}$ does not follow a power-law as a function of temperature. The results are discussed in the context of possible quantum critical nematic fluctuations.

The experimental work is of very high quality. It is very thorough, including also a detailed error analysis, and the manuscript is well written. The topic of possible quantum criticality in these high-temperature superconductors is timely and important, and it is also important to test the here-presented approach. As the authors admit, it is however challenging to explain the obtained results, in particular the absence of a power-law temperature dependence of $m_{B_{2g}}$. This is of course somewhat irritating. It might indeed take a concerted effort of the field to solve the question of the exact temperature and doping dependence of the nematic susceptibility in these systems. Therefore, I think it is important to highlight the experimental results shown here. Indeed, given the frequency with which a Curie-Weiss law is fitted to this kind of data in the literature, these results are an important addition to the discussion in the field. Therefore, I support, in principle, publication of the present manuscript in Nature Communications.

I have two concerns, however.

(1) The error bars and scatter of the data for the temperature dependence of $m_{B_{2g}}$ (Fig. 4a) are much smaller than for the doping dependence of the magnitude of $m_{B_{2g}}$ (Fig. 3a). Furthermore, the deviation from power-law behavior in Fig. 4a, though clear in the data, is rather subtle. To which degree can the authors exclude that a similarly small deviation of power-law behavior is simply hidden in the scatter of the data in Fig.3(a)?

(2) For strongly underdoped compositions, $m_{B_{2g}}$ does indeed follow nicely the Curie-Weiss law, as shown in Fig. 4(b). However, it seems from Fig. 1(b) that slightly underdoped compositions also deviate from a Curie-Weiss-like temperature dependence. If this is correct, what are the implications? I noted that in the underdoped region, the same group has obtained experimental evidence for quantum criticality with a different approach, Ref. [17].

Finally, as a suggestion, it would be useful to make Fig. 3 of the main manuscript and Fig. 4 of the Supplement easier to compare (same axes) and to include the linear fits from which the value of γ was extracted. Can you also add the temperature values at which data points are presented in Fig. 4 (supplement)?

Reviewer #3 (Remarks to the Author):

The present paper deals with power-law behavior of the nematic susceptibility as a function of doping and temperature close to the nematic quantum critical point in $\text{Ba}(\text{Fe}_{1-x}\text{Co}_x)_2\text{As}_2$. I cannot provide positive comments because of many concerns about the present manuscript.

The authors emphasize the presence of the nematic quantum critical point. However, the power law itself is expected close to both quantum and classical critical points. It is not clear why the authors can emphasize much the quantum criticality on the basis of the present measurements. In fact, no evidence is obtained for $\chi \propto T^{-\gamma/z\nu}$ at $x=x_c$.

The authors do not care about a possible role of class fluctuations although their measurements are performed at $T=13$ K and a finite T critical point is indeed present near $x=x_c$ where $T_c(x)$ has a large slope.

The authors emphasize much the Curie-Weiss behavior, which describes mean-field behavior. However, the authors wish to discuss quantum criticality as a function of T . The manuscript should have been formulated in a way that non-experts can easily follow the logical flow of the present work.

It is not clear what kind of nematic fluctuations the authors have in mind. I first guess that it should be Ising nematic fluctuations. However, this sharply contradicts with their data. In the case of 2D Ising universality class, γ is $7/4$ at a finite T critical point and becomes 1 at a quantum critical point. However, the authors obtain $\gamma=0.72$ at $T=13$ K and speculate that γ would become 1 at $T=0$, consistent with the gaussian fixed point; one indeed obtains the gaussian fixed point for the Ising nematic order at $T=0$. However, if so, one would expect γ around $7/4$ close to a finite T critical point and γ would become 1 at a nematic QCP. This tendency is not obtained in the authors' measurements. Rather the opposite tendency is obtained. See Fig. 3 c.

In addition, it seems odd that a critical exponent γ depends on temperature.

In $x>x_c$ ($x=0.068$), the authors conclude that the Curie-Weiss behavior is broken and no single power law is obtained. Given $0.065 < x_c < 0.069$, I would consider $x=0.068$ is very close to the true x_c , not in $x > x_c$. Hence one would expect power-law behavior or some crossover behavior if the authors actually measure critical behavior.

According to Ref. 18, the susceptibility Eq. (1) is a purely electronic one and does not include effects of coupling to the lattice. The actual system, however, contains the effect of coupling to the lattice. Hence the divergence of this susceptibility should not occur at the actual nematic transition in real systems (T^* becomes smaller than T_s in Ref. 18). Now in this manuscript the authors discuss a possible divergence of this susceptibility at the actual QCP in $\text{Ba}(\text{Fe}_{1-x}\text{Co}_x)_2\text{As}_2$ and try to extract a critical exponent. Given that a critical exponent is defined in the vicinity of the critical point and the authors cannot access in principle to the true critical point by using the susceptibility Eq. (1) according to Ref. 18, I wonder how one can safely validate the analysis given in the present manuscript. No consideration is given. In fact, the obtained conclusions are not easily accepted by reflecting on the wisdom of criticality including quantum one.

In the context of the pnictide physics, superconductivity mediated by nematic fluctuations was studied in Y. Yamase and R. Zeyher, Phys. Rev. B 88, 180502(R) (2013).

On page 1, left column, bottom, $\text{Ba}(\text{Fe}_{1-x}\text{P}_x)_2\text{As}_2$ should be corrected.

Reviewer remarks in blue

Our response in red

Reviewer #1 (Remarks to the Author):

I have reviewed the manuscript:

“Comparison of temperature and doping dependence of nematic susceptibility near a putative nematic quantum critical point” by J. C. Palmstrom et al..

This work reports elasto-resistivity measurements in order to follow the temperature dependence of the nematic susceptibility in the vicinity of a putative nematic critical point, x_c , in $\text{Ba}_2(\text{Fe}_{1-x}\text{Co}_x)\text{As}_2$. In order to access the putative nematic critical point, the work reports challenging experiments in 45T as well as in 11T to suppress superconductivity using a large number of samples with compositions around that point. This work accesses the low temperature dependence of the nematic susceptibility and assesses the power law dependences in the vicinity of the x_c both as a function of temperature and composition. It is found that the temperature dependence of the nematic susceptibility does not follow a power law close to x_c whereas a power law is determined as a function of composition. This study brings interesting insight into the nature of quantum criticality in $\text{Ba}_2(\text{Fe}_{1-x}\text{Co}_x)\text{As}_2$.

We respond: We thank the reviewer for their careful reading of our manuscript and interest in our results.

Below I suggest additional aspects to be addressed and clarified:

1) Fig. 1a shows the nematic susceptibility in magnetic field of 11T and 45T and the curves seem to match. The authors could also add to this graph the zero field nematic susceptibility for the same composition $x \sim 0.068$. It is rather surprising that there is no magnetic field effect on the nematic susceptibility that is related to the anisotropy of in-plane resistivity. The magnetotransport and Hall effect of iron-based superconductors have usually non-linear field dependencies due to the multi-band effects and anisotropic scattering. The authors should comment on how exactly these effects cancel out in the nematic susceptibility and an example in the Supplemental Materials would be useful.

We respond: The experiment was performed in the hybrid superconducting magnet at the NHMFL DC Facility. This magnet consists of a static 11.4T superconducting outsert magnet so it was not possible to perform zero field measurements. In the supplements, we now include data on $x \sim 0.079$ taken during an initial field ramp from 0T-45T so a direct comparison with the zero field elasto-resistivity response can be made. There is no discernible field-dependence for any of the measured composition in this field range.

A field independent elasto-resistivity response, even for a field dependent relaxation time τ , is generically allowed since we measure only the (normalized) anisotropy in τ (as a function of field) but not the field dependence of the relaxation time itself. Meaning, we measure the field dependence of the normalized strain derivative of the resistivity, which is distinct from the field dependence of the resistivity. The elasto-resistivity response and appropriate normalization has been derived in previous works (Physical Review B 92, 235147 (2015)).

Furthermore, for this experiment m_{B2g} was determined from the difference of two longitudinal resistance measurements. Therefore, by symmetry, the measurement is insensitive to the Hall effect (which manifests as a transverse voltage). It is sensitive to the magnetoresistance, however, in these materials the magnetoresistance is small and on the order of the error in our ability to measure the elastoresistivity response. We now also include representative data on the magnetoresistance in the supplemental materials.

2) Fig. 1b shows m_{B2g} versus temperature for different composition combining measurements performed in different magnetic fields. The authors should show the related m_{B2g} vs T data in zero field in SM. How the Weiss temperature differs between zero field and the 45T data?

We respond: It is not possible to directly compare the Weiss temperature determined from fits to the data that are taken at zero/low fields and the data that are taken in 45T. The reason is that data obtained in 45T only span a relatively small range of temperatures. Due to the power consumed maintaining 45T fields, the time at these extreme fields is very limited. The 45T data, as shown for example in Fig. 1A, are typically limited between base temperature and just above the zero-field superconducting transition. This is an insufficient temperature range to meaningfully fit the high field data alone. All low field (0T or 11.4T) data are currently already shown in Fig 1B. We reiterate that neither this nor prior studies (Physical Review B 100, 125147 (2019)) have observed a field dependence of the elastoresistivity, and hence the data taken in 45T can be directly compared to the zero-field data, extending the range of those data to much lower temperatures.

3) As the Weiss temperature has been used as an indication of the location of the putative nematic critical point in other studies in different families of iron-based superconductors, it would be useful to have an additional panel in Fig 3 to show the behaviour of the Weiss temperature versus composition and comment on its dependence on composition.

We respond: One of the main findings of our work is that a single power law, such as the Curie-Weiss fit used to extract a Weiss temperature, does not well describe the temperature dependence of the data for heavily doped compositions, far from that of pristine BaFe_2As_2 . While a Curie-Weiss fit may well fit a narrow temperature window for these more highly doped compositions (as has been done previously), the extracted Weiss temperature for compositions close to the putative critical point becomes extremely sensitive to the choice of temperature window. Therefore, for these optimal and overdoped compositions of Co-doped BaFe_2As_2 , a unique Weiss temperature cannot be unambiguously extracted. Indeed, one of our main findings is that for optimal doped compositions, no single power law describes the elastoresistance over a wide temperature range.

4) The findings of this work can be compared to the behaviour of resistivity for the same compositions. The authors could show the resistivity curves in 0T and 45T as separate panels together with their derivative in Figure SM1. This could help to assess the behaviour of resistivity close to x_c , as compared with other studies in the vicinity of a nematic or a magnetic critical point.

We respond: We thank the review for their suggestion and have added a comparison of the resistivity traces in 0T or 11.4T and 45T in Figure SM1.

5) This work identifies a power law dependence between m_{B2g} and the $(x-x_c)^\gamma$ from the overdoped close to x_c at 0.067. What is the behaviour expected from the underdoped side if x_c is the nematic critical point? How this current study compared with the previous findings in 65T published in Phys. Rev. B 100, 125147 (2019)?

We respond: A divergence of the susceptibility upon approach to the critical doping x_c is also anticipated from the underdoped side in the tetragonal phase, however low temperature measurements are precluded due to the finite temperature structural transition and associated domain formation – this is the primary motivation for performing these measurements for overdoped compositions.

Our current study is in good agreement with the findings published in Phys. Rev. B 100, 125147 (2019). In that work, the elastoresistivity was found to be independent of magnetic field up to 65 T and to monotonically increase with decreasing temperature. Due to the limitations of measuring in pulsed field, the elastoresistivity response was not symmetry-decomposed so a direct comparison of the temperature dependence and magnitude is precluded. We emphasize that in the present study, all data sets are fully symmetry-decomposed in order to unambiguously extract the antisymmetric (B_{2g}) response.

6) A temperature of $T \sim 13$ K is used to compare the absolute values of the m_{B2g} as a function of composition the power law dependence on composition in the vicinity of x_c . Can the authors comment on the extrapolated zero temperature values of m_{2g} (0K) versus composition?

We respond: While the magnitude of m_{B2g} appears to monotonically increase with decreasing temperature, we are hesitant to extrapolate to temperatures too far below the measured values because the temperature dependence does not follow a clear functional form. Furthermore, if the elastoresistivity is indeed converging on power-law behavior then it would presumably be appropriate to think of the extrapolation in terms of decades in temperature. In such a scenario, the extrapolation from 13K to 1K is roughly one decade in reduced temperature – approximately equal to the current span of the temperature data set – and, of course, it would require an infinite extrapolation to reach 0K on such a logarithmic scale. Our manuscript therefore emphasizes the observed behavior.

7) In what way the magnetic field play any role in helping to distinguish between a nematic or magnetic quantum critical point? Can the two potential critical points be evaluated from the resistivity data in 45T?

We respond: Our resistivity data (Supplementary Fig. 2) indicate that the magnetic and nematic phase transitions remain separated in the presence of a strong magnetic field down to our base temperature of approximately 4 K, with signatures in the resistivity of only one phase transition for compositions $x=0.0606$, 0.0616, 0.0631, and 0.0648 shown in Supplementary Fig. 2. These signatures smoothly connect to the nematic phase transitions observed for lower compositions -

- compositions for which both the nematic and magnetic phase transitions are observed. We emphasize, though, that our discussion does not rely on identifying the symmetries broken at this putative quantum phase transition. Rather, we identify its location (x_c) and examine the divergence of the elastoresistivity upon approaching it, both as a function of composition and as a function of temperature.

It is sometimes possible to infer the composition at which a QCP occurs by considering the T-dependence of the resistivity, having suppressed the superconductivity with a magnetic field. To do so, it is necessary to measure the resistivity as a function of field at many temperatures, in order to extract the zero-field value of the resistivity (by fitting for fields above H_{c2} , and extrapolating back to zero field for each temperature). This is unfortunately not possible from the present data sets, which focused on T-sweeps at fixed field. Consequently, we estimate the location of the putative QCP as described in the main text (see also our response to Referee 2).

8) What is the role of disorder in understanding the deviation from Curie-Weiss law at low temperatures? In a previous report, the authors suggested that m_{66} coefficient of optimally doped $\text{BaFe}_2(\text{As}_{0.68}\text{P}_{0.32})_2$ follows a perfect Curie-Weiss temperature dependence from $T = 250$ K (Science 352, 6288 (2016)). For $\text{FeTe}_{0.6}\text{Se}_{0.4}$, the m_{66} coefficient was found to be described by the Curie-Weiss dependence “down to T_c but substantially deviates at temperatures greater than 100 K, possibly related to the loss of quasiparticle coherence as a result of its extremely small Fermi energy”.

In what way the Curie-Weiss dependence of nematic susceptibility is sensitive to the details of each system, including the size of the Fermi surface, orbital-dependent effects and disorder?

We respond: Our manuscript emphasizes the experimental observation of a power law divergence of the elastoresistivity coefficient as a function of composition, coupled to a non-power-law divergence with respect to temperature upon approaching the putative QCP. We believe these observations to be significant, in large part because there isn't a simple model that predicts this result, and hence we believe that it should stimulate theoretical discussion.

To address the referee's specific question, we note that disorder in the form of chemical substitution necessarily yields local strains. These strains act as random fields (both longitudinal and transverse) to the Ising nematic order, and are generally expected to suppress the Curie-Weiss divergence in a system. We refer the reviewer to the supplemental information in the Science 352, 6288 (2016) paper for a more in-depth theoretical analysis. The physical origin of the nematic fluctuations (whether driven by orbital or spin effects for instance) does not affect the T-dependence, which in mean-field follow a Curie-Weiss functional form in the absence of disorder.

Of course, we measure something different, which is the elastoresistivity coefficient m_{B2g} , which is only proportional to the $B2g$ nematic susceptibility. This is a transport coefficient, and microscopic details (including band structure details such as the shape/character of the Fermi surface) can/will affect the magnitude, and in principle possibly also the temperature dependence. The observation of Curie-Weiss behavior for underdoped compositions, similar to what is found from thermodynamic probes, is strong evidence that there is not a strong T-dependence to the proportionality coefficient ($g_{t,x}$ in our manuscript) for those compositions. Thermodynamic probes also provide compelling evidence for a deviation from Curie-Weiss behavior for optimal and overdoped compositions (PRL 112, 047001 (2014)). We are, however,

careful in our manuscript to emphasize that we are measuring a transport coefficient, and report its empirical evolution as the putative QCP is approached.

Reviewer #2 (Remarks to the Author):

The authors have experimentally measured and discuss the B_{2g} elastoresistance coefficient (expected to be proportional to the nematic susceptibility) of overdoped Ba(Fe,Co)₂As₂. Superconductivity has been suppressed in magnetic fields up to 45T. The authors show power-law behavior of the magnitude of m_{B2g} at a constant temperature as a function of composition. They also demonstrate that m_{B2g} does not follow a power-law as a function of temperature. The results are discussed in the context of possible quantum critical nematic fluctuations.

The experimental work is of very high quality. It is very thorough, including also a detailed error analysis, and the manuscript is well written. The topic of possible quantum criticality in these high-temperature superconductors is timely and important, and it is also important to test the here-presented approach. As the authors admit, it is however challenging to explain the obtained results, in particular the absence of a power-law temperature dependence of m_{B2g} . This is of course somewhat irritating. It might indeed take a concerted effort of the field to solve the question of the exact temperature and doping dependence of the nematic susceptibility in these systems. Therefore, I think it is important to highlight the experimental results shown here. Indeed, given the frequency with which a Curie-Weiss law is fitted to this kind of data in the literature, these results are an important addition to the discussion in the field. Therefore, I support, in principle, publication of the present manuscript in Nature Communications.

We respond: We thank the reviewer for their thorough reading of our manuscript.

I have two concerns, however.

(1) The error bars and scatter of the data for the temperature dependence of m_{B2g} (Fig. 4a) are much smaller than for the doping dependence of the magnitude of m_{B2g} (Fig. 3a). Furthermore, the deviation from power-law behavior in Fig. 4a, though clear in the data, is rather subtle. To which degree can the authors exclude that a similarly small deviation of power-law behavior is simply hidden in the scatter of the data in Fig.3(a)?

We respond: This is an important point. While the larger error bars on the doping dependence of m_{B2g} do limit the constraint we can put on the functional form, the extent of the deviation from power law seen in the temperature dependence of m_{B2g} can be excluded. This can be shown by fitting the temperature dependence data with artificially increased x and y error comparable to the upper bound of errors in the compositional dependence and comparing the goodness of fit to power law behavior between the compositional and temperature dependences. The resulting reduced χ^2 error in the compositional dependence is 0.94, a value close to one which indicates that the fit, to within error, well describes the data. The reduced χ^2 error of the full temperature dependence is 1.92, a value greater than one which indicates the model does not fully describe the behavior of the data. To illustrate this in the paper, we have added a

comparison figure to the supplemental materials overlaying the temperature and doping dependence of m_{B2g} where it is easy to visually see the deviations from power-law as a function of temperature are greater than the error within the compositional dependence.

(2) For strongly underdoped compositions, m_{B2g} does indeed follow nicely the Curie-Weiss law, as shown in Fig. 4(b). However, it seems from Fig. 1(b) that slightly underdoped compositions also deviate from a Curie-Weiss-like temperature dependence. If this is correct, what are the implications? I noted that in the underdoped region, the same group has obtained experimental evidence for quantum criticality with a different approach, Ref. [17].

We respond: The measurements presented in Ref. [17] which demonstrate the presence of quantum critical nematic fluctuations in the underdoped regime for this material system were performed above the zero-field superconducting dome, i.e. at or above 25 K. Since the measurements in this work are closer both in temperature and doping to the putative quantum critical point, one might naively assume that clear power law scaling of the elastoresistivity might also be observed in this regime. However, the present work is performed for overdoped compositions, and hence a direct comparison is unfortunately not possible. The sub-Curie Weiss behavior seen in the slightly underdoped samples does, however, suggest that the system is sensitive to the proximity to the putative critical point. This could be due to either the behavior of $g_{t,x}$ close to a critical point or because the sensitivity to disorder is generally considered to be greater closer to a quantum critical point. The latter implies that there might be a second energy scale, potentially related to disorder in the system, that plays a role at low temperatures.

Finally, as a suggestion, it would be useful to make Fig. 3 of the main manuscript and Fig. 4 of the Supplement easier to compare (same axes) and to include the linear fits from which the value of γ was extracted. Can you also add the temperature values at which data points are presented in Fig. 4 (supplement)?

We respond: Thank you for the suggestion. We have updated the axes on the supplemental Fig. 4 to match Fig. 3 of the main text and have added the linear fits used to extract the power law exponents. Due to the density of data presented in the supplemental Fig. 4 it is not practical to add individual temperature labels for every trace, however there is a colorbar that maps the color scheme to temperature.

Reviewer #3 (Remarks to the Author):

The present paper deals with power-law behavior of the nematic susceptibility as a function of doping and temperature close to the nematic quantum critical point in $\text{Ba}(\text{Fe}_{1-x}\text{Co}_x)_2\text{As}_2$. I cannot provide positive comments because of many concerns about the present manuscript.

The authors emphasize the presence of the nematic quantum critical point. However, the power law itself is expected close to both quantum and classical critical points. It is not clear why the authors can emphasize much the quantum criticality on the basis of the present measurements. In fact, no evidence is obtained for $\chi \propto T^{-\gamma/z}$ at $x=x_c$.

We respond:

As stated in the introduction of our paper, there is both experimental and theoretical evidence that supports the presence of a quantum critical point in the Fe-based superconductors. This evidence is not conclusive, but strongly motivates comparing the measured elastoresistivity response, a proxy for the electronic nematic susceptibility, with the expected behavior near a possible quantum critical point. We emphasize that our work primarily presents experimental results. These results are analyzed in such a way as to be directly comparable with current theoretical models, however, we do not claim that we present direct evidence for a quantum critical point in this system nor, to our knowledge, do any of the current theoretical models describe the observed temperature and doping dependence of the elastoresistivity. Rather, we present new experimental observations that we believe present a clear picture of the elastoresistivity response that diverges upon approach to the putative QCP as a function of composition from the underdoped side of the phase diagram.

We agree with the reviewer that the main conclusion from our data is that the observed elastoresistivity response cannot be described by the expected behavior on approach to a clean nematic quantum critical point. This is indeed the main result of our paper.

The authors do not care about a possible role of class fluctuations although their measurements are performed at $T=13$ K and a finite T critical point is indeed present near $x=x_c$ where $T_c(x)$ has a large slope.

We respond: Classical fluctuations diverge at a critical temperature $T_c(x)$, while quantum critical fluctuations are scaled by a critical value of a nonthermal tuning parameter, such as doping (i.e. at x_c). These two scenarios can be differentiated by the evolution of the susceptibility as a function of doping. As can be seen from the data, i.e. Fig. 3a, the data is well scaled by a critical doping. This power law behavior spans almost a decade in reduced composition, including far overdoped compositions far from the finite temperature classical transition.

The authors emphasize much the Curie-Weiss behavior, which describes mean-field behavior. However, the authors wish to discuss quantum criticality as a function of T . The manuscript should have been formulated in a way that non-experts can easily follow the logical flow of the present work.

We respond: The discussion with respect to the Curie-Weiss functional form is very important in the context of these materials and this measurement technique. The Curie-Weiss functional form has historically been used to fit the elastoresistivity response in this and other systems. Therefore the deviation from Curie-Weiss behavior for measurements over an extended temperature range is an important consideration for all future analysis.

It is not clear what kind of nematic fluctuations the authors have in mind. I first guess that it should be Ising nematic fluctuations. However, this sharply contradicts with their data. In the case of 2D Ising universality class, γ is $7/4$ at a finite T critical point and becomes 1 at a quantum critical point. However, the authors obtain $\gamma=0.72$ at $T=13$ K and speculate that γ would become 1 at $T=0$, consistent with the gaussian fixed point; one indeed obtains the gaussian fixed point for the Ising nematic order at $T=0$. However, if so, one would expect γ around $7/4$ close to a finite T critical point and γ would become 1 at a nematic

QCP. This tendency is not obtained in the authors' measurements. Rather the opposite tendency is obtained. See Fig. 3 c.

In addition, it seems odd that a critical exponent γ depends on temperature.

We respond: As the reviewer has previously pointed out, the data are not consistent with the expected behavior on approach to a clean quantum critical point. To our knowledge no theoretical model is able to explain the observed behavior. Hence, we do not attempt to assign the extracted exponents to any theoretical model or universality class. We note that we do ***not*** speculate the zero-temperature value of γ to be 1 nor do we assign its behavior to a gaussian fixed point. We have removed our reference to the expected values of the exponent for mean-field and Hertz-Millis critical points as it appears to have caused some confusion.

Just to clarify an important point for the referee, although this goes beyond the discussion in the paper, the material is anticipated to fall in to the universality class of 3d Ising with long range interactions. The long range strain interactions yield an upper critical dimension of 2, and hence mean field behavior is anticipated in the absence of disorder (see, for example, U Karahasanovic & J Schmalian, "Elastic coupling and spin-driven nematicity in iron-based superconductors" PRB 93, 064520 (2016).). Of course, disorder changes this analysis, while coupling of the critical nematic fluctuations to the Fermi surface can also modify the dynamical scaling exponent.

In $x > x_c$ ($x=0.068$), the authors conclude that the Curie-Weiss behavior is broken and no single power law is obtained. Given $0.065 < x_c < 0.069$, I would consider $x=0.068$ is very close to the true x_c , not in $x > x_c$. Hence one would expect power-law behavior or some crossover behavior if the authors actually measure critical behavior.

We respond: $x=0.068$ was indeed chosen as the representative sample since it was the nearest sample to the putative critical point, and thus the most likely to show the expected power law behavior if such behavior existed in this system. We have replaced $x=0.068$ greater than or approximately x_c with $x=0.068 \approx x_c$. The previous notation was meant to emphasize our uncertainty in x_c , not the distance of 0.068 from the critical point.

According to Ref. 18, the susceptibility Eq. (1) is a purely electronic one and does not include effects of coupling to the lattice. The actual system, however, contains the effect of coupling to the lattice. Hence the divergence of this susceptibility should not occur at the actual nematic transition in real systems (T^* becomes smaller than T_s in Ref. 18). Now in this manuscript the authors discuss a possible divergence of this susceptibility at the actual QCP in $\text{Ba}(\text{Fe}_{1-x}\text{Co}_x)_2\text{As}_2$ and try to extract a critical exponent. Given that a critical exponent is defined in the vicinity of the critical point and the authors cannot access in principle to the true critical point by using the susceptibility Eq. (1) according to Ref. 18, I wonder how one can safely validate the analysis given in the present manuscript. No consideration is given. In fact, the obtained conclusions are not easily accepted by reflecting on the wisdom of criticality including quantum one.

We respond: This is a very important point which deserves discussion in the manuscript. We have updated the main text and supplemental information to include this.

By measuring the strain experienced by the material, elastoresistivity measurements probe the 'bare' electronic nematic fluctuations of the system. Meaning, the measured nematic susceptibility is purely electronic in origin and independent of any cooperative effects from lattice softening. The advantage to this methodology is that the critical exponent and scaling of the measured electronic nematic susceptibility is well defined, even in the presence of phonon softening. Therefore, this becomes a question of only the choice of critical doping for the power law scaling ($m \sim |x - x_c|^\phi$).

The reviewer is indeed correct that the structural transition includes effects from coupling to the lattice and the electronic nematic susceptibility should diverge at the 'bare' electronic nematic quantum critical point (x_{c_bare}), if such a point exists in this system. In practice, we cannot determine x_{c_bare} for the Co system since the elastoresistivity response does not follow a known temperature dependence. However, the cleaner P-BaFe₂As₂ system follows a Curie-Weiss functional form near optimal doping. In this system the Weiss temperature was suppressed towards zero near optimal doping and close to where the structural transition is also suppressed towards zero. Therefore, we make the assumption in this work that $x_{c_bare} \approx x_c$.

We also now include in the supplemental information power law fits of m_{b2g} for extremal values of x_{c_bare} , far from the physically motivated values of the critical point where the elastoresistivity follows a Curie-Weiss functional form with a Weiss temperature well above zero). We show that the doping dependence is still well fit by a power law functional form and thus the main conclusions of our paper still qualitatively hold even for a large range of potential values of the critical doping.

In the context of the pnictide physics, superconductivity mediated by nematic fluctuations was studied in Y. Yamase and R. Zeyher, Phys. Rev. B 88, 180502(R) (2013).

We respond: We have added this reference to the discussion on the theoretical link between nematic fluctuations and superconductivity.

On page 1, left column, bottom, Ba(Fe_{{1-x}P_x})₂As₂ should be corrected.

We respond: Thank you for the correction. We have corrected this to BaFe₂(As_{{1-x}P_x})₂

REVIEWER COMMENTS

Reviewer #1 (Remarks to the Author):

The authors have addressed all the Referees' comments and made appropriate changes to their manuscript.

Reviewer #2 (Remarks to the Author):

The authors have responded to all the criticisms of the referees in detail and, in many cases, with additional data and analysis added to the supplement. Overall, I am satisfied with their explanations and support publication of the manuscript.

I would like to point out one issue, though: I put in some effort to "to visually see the deviations from power-law as a function of temperature are greater than the error within the compositional dependence" in the additional supplementary figure 9. I do not consider this "easy" to see. Whereas it may be acceptable to say that the compositional data fall better on a straight line than on the curved line of the temperature dependence, this is not very clear. For example, changing the compositional scale to the range 0.004-0.4, the distinction becomes hard to make. The comparison of the goodness of fit is a clearer criterion. The authors' summarizing statement: "elastoresistivity [...] diverges as a function of composition as x approaches x_c from the overdoped side of the phase diagram, with a functional form that is not markedly inconsistent with a simple power law ... ", captures this accurately, though.

Reviewer #3 (Remarks to the Author):

The authors revise the manuscript very carefully and the revised one is now written very well. However, it is not clear how important the authors' findings are. The exponent of ϕ , which is the same as γ in the vicinity of a critical point, depends on temperature and is not universal. This is not easily understood in the existing knowledge. In addition, the temperature dependence of m_{B2g}^{B2g} does not follow a power law at least down to 13 K at $x=x_c$ in spite of the presence of a putative nematic QCP. The physical implication of this behavior is not clear. Crucially those two findings are not harmony with the principle of criticality. I am afraid that the authors fail to pick up true critical fluctuations. In fact, there are experimental limitations that the lowest temperature is 13 K in the present work and the clear understanding of $g_{T,x}$ in Eq. (1) has not been obtained in $x > x_c$. In addition, there is ambiguity about a choice of x_c . Hence the scientific messages from the present work become rather tentative unfortunately. I would refrain from recommending publication of the present work in Nature Communications.

On top of the above two fundamental concerns, I still frown on the obtained value of $\phi=0.72$, which is not understandable even in 3D Ising universality class (γ approx 1.24). Although a disorder effect may modify the value of γ , there are no convincing arguments to support such a drastic reduction from 1.24 to 0.72. Rather as I mention above, I would worry about the experimental limitation. A choice of x_c does matter to estimate a value of γ ; see the supplementary information. To get a reasonable γ , the authors may assume a smaller x_c . However, this yields an additional problem that m_{B2g}^{B2g} is suppressed at low T in $x < 0.067$ because of the structural transition, preventing the authors from analyzing nematic QCP behavior in a compelling way in terms of m_{B2g}^{B2g} .

Reviewer remarks in blue

Our response in red

Reviewer #1 (Remarks to the Author):

The authors have addressed all the Referees' comments and made appropriate changes to their manuscript.

We thank the reviewer for both their time and effort invested in reviewing our manuscript and their support for publication.

Reviewer #2 (Remarks to the Author):

The authors have responded to all the criticisms of the referees in detail and, in many cases, with additional data and analysis added to the supplement. Overall, I am satisfied with their explanations and support publication of the manuscript.

We thank the reviewer for their support of our manuscript and their critiques which have helped bring the manuscript to its current form.

I would like to point out one issue, though: I put in some effort to "to visually see the deviations from power-law as a function of temperature are greater than the error within the compositional dependence" in the additional supplementary figure 9. I do not consider this "easy" to see. Whereas it may be acceptable to say that the compositional data fall better on a straight line than on the curved line of the temperature dependence, this is not very clear. For example, changing the compositional scale to the range 0.004-0.4, the distinction becomes hard to make. The comparison of the goodness of fit is a clearer criterion. The authors' summarizing statement: "elastoresistivity [...] diverges as a function of composition as x approaches x_c from the overdoped side of the phase diagram, with a functional form that is not markedly inconsistent with a simple power law ... ", captures this accurately, though.

This is a fair point and our choice of phrasing in our previous response was poor. This phrasing and argument were only used in our written response to you. Both the main text and supplemental material rely only on the goodness of fit criterion so the manuscript remains unchanged.

Reviewer #3 (Remarks to the Author):

The authors revise the manuscript very carefully and the revised one is now written very well.

We are glad that our current iteration is clearer. Thank you for your previous comments pointing out sections which were confusing.

However, it is not clear how important the authors' findings are. The exponent of ϕ , which is the same as γ in the vicinity of a critical point, depends on temperature and is not universal. This is not easily understood in the existing knowledge.

This is actually not unprecedented in the presence of disorder. For instance, rare-region effects can introduce a temperature dependence to the scaling exponent z . This has recently been

observed in another iron-based system, FeSe_{1-x}S_x (arXiv:2103.07991, now Ref. 27). We have added discussion to this effect to the paper.

In addition, the temperature dependence of $m_{\{B2g\}^{\{B2g\}}}$ does not follow a power law at least down to 13 K at $x=x_c$ in spite of the presence of a putative nematic QCP. The physical implication of this behavior is not clear. Crucially those two findings are not harmony with the principle of criticality. I am afraid that the authors fail to pick up true critical fluctuations. In fact, there are experimental limitations that the lowest temperature is 13 K in the present work and the clear understanding of $g_{\{T,x\}}$ in Eq. (1) has not been obtained in $x > x_c$. In addition, there is ambiguity about a choice of x_c . Hence the scientific messages from the present work become rather tentative unfortunately. I would refrain from recommending publication of the present work in Nature Communications.

Respectively, we disagree with these criticisms. Our work presents an empirical study of the behavior of the nematic fluctuations in a real system with finite disorder. These measurements were performed in temperature ranges where signatures of nematic quantum criticality have already been observed. The fact that the behavior of the system differs from the expected theoretical behavior is in itself a meaningful result as it points out a fundamental challenge to understanding these systems.

Experimental observations of nematic quantum criticality in this system at temperatures ranges significantly above the lowest temperature range in this work can be found in Ref. 9 and arXiv:2111.07521 (now Ref. 10). In addition, we note that for samples close to x_c , $g_{\{T,x\}}$ has been shown to be temperature independent (Ref. 19), indicating that the deviation from power law as a function of temperature of $m_{\{B2g\}^{\{B2g\}}}$ is a robust intrinsic property and not an experimental artifact.

On top of the above two fundamental concerns, I still frown on the obtained value of $\phi=0.72$, which is not understandable even in 3D Ising universality class (γ approx 1.24). Although a disorder effect may modify the value of γ , there are no convincing arguments to support such a drastic reduction from 1.24 to 0.72. Rather as I mention above, I would worry about the experimental limitation. A choice of x_c does matter to estimate a value of γ ; see the supplementary information. To get a reasonable γ , the authors may assume a smaller x_c . However, this yields an additional problem that $m_{\{B2g\}^{\{B2g\}}}$ is suppressed at low T in $x < 0.067$ because of the structural transition, preventing the authors from analyzing nematic QCP behavior in a compelling way in terms of $m_{\{B2g\}^{\{B2g\}}}$.

For a nematic quantum critical point in a metal, the system is anticipated to be in a regime where $d + z > 4$. This is due in part to the finite nemato-elastic coupling. Upon approach to the critical point the C₆₆ component of the elastic tensor softens, resulting in an anisotropic correlation length and an angular dependent mass of the nematic fluctuations. A consequence of this is that the effective dimensionality of the system is enhanced such that the system is above the upper critical dimension for both the thermal and quantum phase transitions. This effect has been extensively discussed the literature: see, for example Ref. 15. The standard Hertz-Millis approach predicts the mean-field exponent, $\gamma = 1$.

That said, we believe that the current theory describing the anticipated behavior near a clean quantum critical point does not fully capture the observed experimental response. This is one of the main conclusions of our work and is a robust observation independent of choice of fit

parameters (i.e. x_c). Furthermore, as detailed in our above response, we believe we are in a temperature regime where such scaling should occur and that the deviations seen in the temperature dependence are not an experimental artifact. The observed power law scaling as a function of composition cannot be interpreted in isolation from the deviations seen as a function of temperature. We therefore present an empirical overview of our experimental results and fits. Since the behavior does not match the theoretical models, we cannot use the exponent as a function of doping (ϕ) to definitively rule out theoretical models by direct comparison with the theoretically defined universal scaling exponent (γ). We note that the behavior of disordered systems can be fundamentally different than their clean counterpart. For instance, rare-regions within a disordered system may introduce a temperature dependence to z . In such a system the associated susceptibility would then deviate from the anticipated power law behavior, reminiscent of the observed temperature dependence of $m_{\{B2g\}}^{\{B2g\}}$ at x_c .